# Empowering Commercial Vehicles through Data-Driven Methodologies

Paolo Bethaz [1,*], Sara Cavaglion [2], Sofia Cricelli [2], Elena Liore [2], Emanuele Manfredi [2], Stefano Salio [3], Andrea Regalia [2], Fabrizio Conicella [3], Salvatore Greco [1] and Tania Cerquitelli [1]

[1] Department of Control and Computer Engineering, Politecnico di Torino, 10129 Turin, Italy; salvatore_greco@polito.it (S.G.); tania.cerquitelli@polito.it (T.C.)
[2] Accenture S.p.A., 10126 Turin, Italy; sara.cavaglion@accenture.com (S.C.); sofia.cricelli@accenture.com (S.C.); elena.liore@accenture.com (E.L.); emanuele.manfredi@accenture.com (E.M.); andrea.regalia@accenture.com (A.R.)
[3] CNH Industrial, 10156 Turin, Italy; stefano.salio@cnhind.com (S.S.); fabrizio.conicella@cnhind.com (F.C.)
[*] Correspondence: paolo.bethaz@polito.it

**Abstract:** In the era of "connected vehicles," i.e., vehicles that generate long data streams during their usage through the telematics on-board device, data-driven methodologies assume a crucial role in creating valuable insights to support the decision-making process effectively. Predictive analytics allows anticipation of vehicle issues and optimized maintenance, reducing the resulting costs. In this paper, we focus on analyzing data collected from heavy trucks during their use, a relevant task for companies due to the high commercial value of the monitored vehicle. The proposed methodology, named TETRAPAC , offers a generalizable approach to estimate vehicle health conditions based on monitored features enriched by innovative key performance indicators. We discussed performance of TETRAPAC in two real-life settings related to trucks. The obtained results in both tasks are promising and able to support the company's decision-making process in the planning of maintenance interventions.

**Keywords:** connected vehicles; predictive maintenance; applied data science; telematics data; interpretable models

## 1. Introduction

In many companies in a wide variety of sectors, data play an increasingly important role. In recent years, there has been a great deal of work demonstrating the benefits obtained from data collection and data analytics operations performed in a precisely targeted way. Among the sectors that can benefit from this, there is undoubtedly the automotive sector, where data-driven strategies can be implemented during the industrial process and in the monitoring of resources during their use. Connected vehicles generate during their use a lot of information through the exploitation of intelligent sensors. The data collected are then analyzed in more detail in a second step to extract useful insights to support the company's decision. This process is particularly well suited in commercial vehicles (trucks/buses) due to their significantly higher cost compared to other vehicles, thus allowing for the possibility to minimize expenses and running costs.

Here we focus on the analysis of telematics data collected on heavy trucks. Little work has been performed so far on this kind of vehicle. Due to their high economic impact, finding benefits from telematically collected data is a challenge of great interest for researchers and practitioners. In this paper, we introduce TETRAPAC (TElematic data of TRucks for Advanced Predictive Analysis of their Component), a general methodology able to manage telematics data collected from trucks, provide an innovative data-driven strategy based on features and key performance indicators (KPIs), and model the vehicle health status. The proposed methodology includes data preprocessing and predictive analytics steps that can

be generalized and tailored to different use cases. Further, TETRAPAC offers in-depth knowledge and interpretability of the results through the definition of innovative KPIs able to add information content to the initial monitored features, allowing the company to exploit the obtained information with greater facility, checking the condition of the vehicle, and scheduling maintenance intervention accordingly, saving costs and time.

We assessed the methodology's performance on real data collected from trucks belonging to a well-known multinational corporation. The evaluation of the methodology has been performed on two different use cases: (1) identifying vehicles with potential DTC (diagnostic trouble code) and (2) estimation of the battery life of the trucks. Both results obtained on the two use cases seem promising and lead to several benefits for stakeholders.

The paper is organized as follows. Section 2 defines the business case within which this work occurs, while Section 3 discusses the state-of-the-art works in the same context, highlighting the differences between these works and the one proposed in this paper. Section 4 details the methodology, from the preprocessing step to the evaluation of the predictive results. Section 5 shows the environment and technologies used for data storage and data analysis. In contrast, Sections 6 and 7 detail the results obtained by using the proposed methodology on two different use cases: the first to predict DTC, the second to estimate the state of the battery. Finally, Section 8 summarizes the content of our work, providing conclusions and suggestions for future improvements.

## 2. Business Case

Nowadays, in the context of connected vehicles, the alliance between data science and business is rapidly growing. Companies constantly monitor their vehicles to make better business decisions based on proper studies on real-time data. Indeed, in connected vehicles, sensors collect telematics data and send them to a black-box installed on-board, which, in turn, forwards them to the cloud service, where they can be analyzed and manipulated. Telematic data monitors both internal and external parameters. Internal parameters monitor the truck's usage (e.g., state of charge of the battery, alert signal on the panel). The second characterizes the mission (e.g., the GPS signals, the environment temperature).

One of the main challenges of automotive companies is represented by *predictive maintenance*, i.e., the possibility of identifying in advance malfunctions or component issues and to promptly intervene before they occur. This aspect is crucial both for economic saving and customer satisfaction. Predictive methodologies allow (i) avoiding breakdowns during vehicle travel, (ii) preventing that neglected alarms of a failure lead to damage other parts, (iii) replacing components only if it is objectively necessary, and (iv) better planning of workshop activities, both for an automotive company and the customer.

We first detail all of them, showing how they are strongly connected and why they are so worthwhile for the company, to then figure out how machine learning algorithms are a valuable solution.

Sudden vehicle breakdowns create customer dissatisfaction and often involve high costs. Customer dissatisfaction occurs because the driver is forced to stop and contact the support center, wasting precious time and feeling vulnerable. High costs are related to vehicle recovery, the substitution of the component that caused the breakdown, and the repair/substitution of other parts, eventually affected by the principal failure. The prediction could support the definition of better scheduled maintenance and cost savings for warranty (e.g., refusing the payment for the repair of components involved by the principal loss if the driver does not consider the advice of stopping in a workshop). The possibility of notifying the driver to stop repairing a component that is wearing down, constantly monitoring their health of status, represents a positive aspect also for the customer, who can optimize their route, plan, and reduce the total number of stops in the workshop.

On the other hand, the customer changes components when their warranty is still valid, even though they are not ruined. This aspect implies high costs to be sustained by

the company. Consequently, the possibility of objectively knowing their health status could build a more vital faithfulness between the two parts.

In this framework, data science methodologies applied to telematics data could lead to achieving good business results. Here we focus on two real-life settings:

**Case 1: Predicting Diagnostic Trouble Codes (DTC)**

The goal is to predict the presence of a DTC ignition in the future. Forecasting the appearance of a DTC allows the company to intervene, avoiding sudden vehicle breakdown, with the consequent economic and time-saving benefits, and a significant and prompt increase in customer satisfaction. The better the model, the lower the number of recalled vehicles (i.e., only trucks that will have a problem should be identified). In this scenario, it is better to stop a vehicle that will present a DTC than calling back one that will not have a DTC ignition since, in the latter case, the driver's mission will be negatively impacted with an increase in customer dissatisfaction. To perform the analysis, trucks with at least a failure and those without losses have been analyzed jointly to allow machine learning algorithms to differentiate among critical and regular missions. Telematic variables have been integrated with the claim's data and analyzed as discussed in Sections 4.1 and 4.2.

Another useful metric to evaluate the case study can be expressed in terms of the probability for each vehicle to present a failure after a specific period of time. Once vehicles are ordered in a descending way based on their probability of failure, different management actions can be performed based on it, planning the right proactive reaction of the control room, managing failure, and the callback of trucks to optimize the overall process and operating associated costs.

**Case 2: Forecasting battery state of health (SOH)**

The aim here is to forecast the battery replacement, thus avoiding unnecessary battery changes when SOH is still high. When the prediction of the battery state of health (ideally a real value from 0 to 100) is correctly performed, it could lead to economic and time-saving benefits and an improvement in customer satisfaction. The main objective here is to minimize the sum of the error (i.e., the differences) between the actual values and the predicted ones.

All vehicles monitored by telematics have been analyzed, and new relevant KPIs have been defined based on domain knowledge. The TETRAPAC methodology (see Section 4.1) and machine learning algorithms (see Section 4.2) are exploited to predict the SOH in a future time window.

### 3. Literature Review

With the advent of the Internet of Things, the exchange of vast amounts of data in real time has become a widespread practice in many scenarios. This has brought particular benefits in the context of Industry 4.0, where, by monitoring the different components present in a production cycle through sensors, it is now possible to have greater control over all stages of production. If appropriately collected and with a precise target, the parameters monitored by the sensors can offer significant benefits to the company. To this aim, the IoT world is empowered with data mining and machine learning techniques that, thanks to their data-driven nature, can highlight important aspects that can guide business decisions, thus offering economic benefits on the business side.

Several works have already been proposed in an Industry 4.0 scenario. The authors of [1] collect a series of works carried out from 2015 to 2020, based on machine learning techniques to facilitate predictive maintenance in manufacturing contexts. For example, both [2,3] propose a data-driven bearing performance degradation assessment method, monitoring bearing running states to ensure machine safety; however, in this paper, we deviate from a purely manufacturing context, seeking to exploit the benefits that IoT has brought to connected vehicles. In connected vehicles, the term Internet of Vehicles (IoV) is used [4,5], referring to the situation in which data can be locally collected on the vehicle and then sent to a remote storage location (cloud) where they can be analyzed in more

detail. In this context, the authors in [6] propose an abstract network model of the IoV, discussing the technologies required to create the IoV; however, the work presented in [6] focuses mainly on technologies that can efficiently collect data from a vehicle without going into detail about techniques linked to the world of big data and machine learning that enable the discovery of helpful information to be extracted from this data. Instead, the authors in [7] analyzed monitored parameters to predict the fuel consumption of a car through the use of a neural network.

The parameters collected telematically on the vehicle can be exploited on the business side mainly for two functions: (i) predictive maintenance and (ii) remaining useful life (RUL) estimation. Regarding the first research challenge, work on this issue has, for example, been presented in [8], which attempts to estimate the probability of fuel pump failure, or in [9], where authors investigated the feasibility of prognostics and health management under different driving circumstances. Instead, the RUL estimation consists of estimating the residual life of a given vehicle (in terms of the number of kilometers it can still cover or days of activity). This information can be beneficial when referring to commercial vehicles, for example, during a buy-back operation, as illustrated in [10]. Work with a similar aim has been proposed in [11], where the authors propose a cloud-based infrastructure capable of providing an online assessment of the residual helpful life and end of life (EOL) of the vehicle under analysis.

Our work differs significantly from those listed above. There are three main innovations introduced here:

- First of all, unlike [7,9], we do not consider a generic connected vehicle case (cars), but we focus on the telematics of trucks. Not much work in this context has been performed on heavy trucks at present, leaving plenty of room for new analyses and experiments. The work presented in [10] is based on a situation similar to the current one; however, the analyses performed are completely different in terms of both the research objective and methodology adopted. Only a portion of the analyzed data is common to the current study.
- Different from a research activity, such as [10], in which the only parameters considered are those monitored by the various sensors, here we enrich the available telematics data with a proper feature modeling step. The definition of specific key performance indicators (KPIs) allows capturing the knowledge hidden in the various parameters being monitored.
- Finally, the work presented in this paper aims to describe a general-purpose methodology tested on two different use cases. The generality feature significantly appreciates in companies, which is another crucial aspect of the proposed approach.

## 4. Methodology

Here we present the TETRAPAC methodology (TElematic data of TRucks for Advanced Predictive Analysis of their Component), whose graphical representation is shown in Figure 1. It combines a solid theoretical background with a necessity to solve real business needs. Leveraging telematics data, TETRAPAC brings tangible benefits both to the company and final customers.

In particular, it consists of a KDD process (knowledge discovery process from data) adapted to business needs: prevent any failure occurrences from optimizing fleet management.

In case (1), the main objective is to detect, with a given time horizon, an error ignition (DTC) through machine learning models, addressing a classification task on telematics data. The main goal, based on business needs, is to reduce false positives and maximize true positives. In this way, we could minimize the number of useless workshop callbacks.

In case (2), the focus is predicting battery health status in an optimal period of time, facing a regression problem on telematics data. Consequently, the analysis is performed to obtain values closest as possible to the real ones.

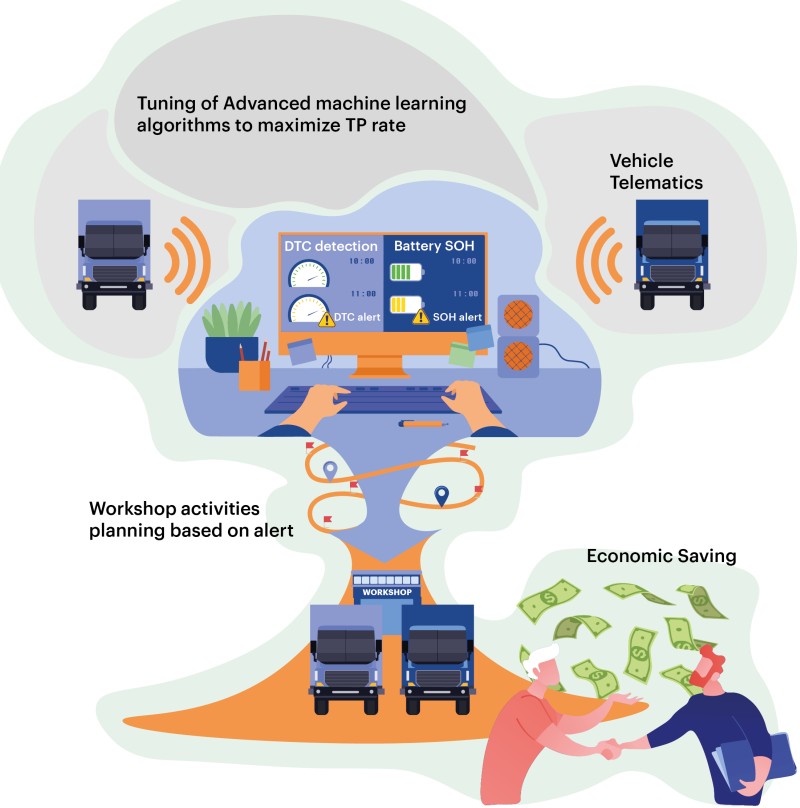

**Figure 1.** The TETRAPAC overview.

TETRAPAC consists of four building blocks, as shown in Figure 2: before real-time deployment, there is a *data preprocessing* step to clean and merge data from different sources, a *predictive analytics* step to derive the most suitable descriptive model to perform accurate predictions, and *evaluation and interpretation of results* to assess the goodness of the model through specific metrics. In the following sub-sections, we reported a detailed description of each building block. The only differences between the two use cases are in the data preprocessing step, in particular in feature modeling, and in the predictive analytics one, since use cases' scopes deal with different types of target variables.

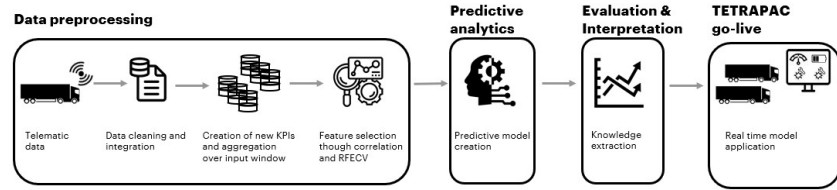

**Figure 2.** TETRAPAC building blocks.

### 4.1. Data Preprocessing

TETRAPAC preprocessing is customized to prepare data in the most suitable way through three specific steps: data cleaning and integration, feature modeling and feature selection.

The above phases are fundamental for the success of the analysis.

The data cleaning and integration step prepares data, providing a good-quality dataset as an input to the model to improve the overall performance: bias and noise must be detected and removed.

Moreover, a variety of data related to trucks is needed to obtain good results; thus, a data integration phase must have a global view of data stored on different sources: telematic transmissions, vehicle registry, and failure collections.

For each vehicle, we extracted all data related to single trips, defined by departure and destination. In case of anomalies, e.g., transmission or estimation errors, the corresponding data were removed. Null values were substituted or eliminated according to their different causes. They were eliminated if they were present in the target variable, otherwise they were replaced to avoid the loss of data to train the model, with the weighted mean, to take into account the different length of travel.

The next step is the feature modeling, whose aim is defining new KPIs and time-independent features to better model the key aspect of data under analysis. This step was performed with the support of domain experts.

- New KPIs creation
  On one side, we define general KPIs (useful for both use cases) concerning key aspects of a vehicle's past life, such as *thermal_excursion* to monitor the external temperature, *days_in_stock* to count the number of days spent in stock, and *days_before_failure* to keep track of time passed before the error occurred. Namely, it monitors the number of days between the first vehicle transmission and the failure. On the other side, for use case (2), we create specific KPIs to keep track of battery charge–discharge cycles. We monitor the *state of health* (SOH) and the *state of charge* (SOC) through, for example, *delta_discharge* to monitor how much battery discharged between two consecutive trips and *charging_speed* to evaluate how fast the charging process is.

- Time-series feature representation
  Aggregation over time has to be treated with attention because we are dealing with *multi-variate discrete time series*. Data should not be flattened to avoid any loss of information, but temporal trends must be tracked to save the information of any specific peaks or behaviors. Our sequential set of data points is represented by variables monitored by telematics data over time before failure. To capture their trend over time effectively, we perform a feature-based time-series representation. In particular, we compute the following statistics and metrics: minimum, maximum, average, standard deviation, percentiles at 25, 50, and 75, discrete integral (obtained from the ratio between the sum of values assumed in a defined window and time elapsed traveling during that period), intercept, and slope through linear regression. Regarding parameters that monitor occurrences, time aggregation consists of dividing the sum of events by time elapsed traveling during that window.

For the feature selection step, we start with the analysis of correlation through Pearson's coefficient to remove correlated features and reduce noise. We addressed two issues: (i) removing parameters highly related to each other—for each couple of features with Pearson's coefficient higher than 0.9 in absolute value, a single variable is considered in the subsequent analytic steps; (ii) removing features uncorrelated with the target variable—the input variable showing a Pearson's coefficient of less than 0.005 or 0.15 in absolute value based on use case framework with the target variable was removed.

Then we exploit a precise procedure: *Recursive Feature Elimination with Cross-Validation* (RFECV) (https://github.com/scikit-learn/scikit-learn/blob/95119c13af77c76e150b753485c662b7c52a41a2/sklearn/feature_selection/_rfe.py (accessed on 1 September 2021)).

It is based on creating a parameter ranking to perform recursive feature elimination and, in the end, it finds the best subset evaluating model performance and cross validating the result. Obtaining a restricted set of top predictors helps to reduce noise in the model and makes result interpretation more straightforward and more effective for business experts.

### 4.2. Predictive Analytics

The entire analytics process includes two phases: (i) offline model building, in which TETRAPAC selects the optimal algorithm and the corresponding input parameter setting, and (ii) online model usage, where predictions are performed in real time on new unseen data.

Since we are dealing with slow degradation phenomena, it is necessary to set the analysis on specific time windows. We perform Feature Modeling in both its steps (KPIs creation and time-series feature representation). A crucial aspect of reaching our purpose of correct failure detection is organizing data in three different time windows, as shown in Figure 3. The model training will exploit data aggregated over the input window. Then, after a period of time equal to the blind window, the algorithm's output will show the predicted occurrence of the failure. Detection of the optimum values for time window sizes is achieved through experiments. We leverage the optimal structure of time-windows in both analytics steps.

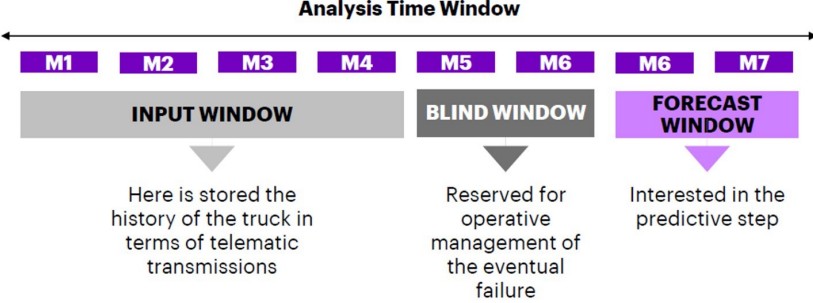

**Figure 3.** Time windows setting.

The model building phase allows creating different kinds of predictive models intending to obtain the best performance. Methods chosen are mainly tree-based, such as *random forest* (for details, see [12]) and *XGBoost* (extreme gradient boosting) [13]. These are two examples of ensemble learning techniques. In particular, random forest combines multiple decision tree models to improve general performance. One of the advantages is that this technique has higher accuracy than single decision trees, and it is also more robust to noise and outliers. Then, the statistical framework of XGBoost casts boosting as a numerical optimization problem where the objective is to minimize model loss by adding weak learners using a stochastic gradient descent-like procedure. Another advantage is that it is parallelizable. Depending on the specific use case proposed, these algorithms will perform regression or classification.

After model creation, we proceed with real-time deployment exploiting telematics data prepared as discussed in Section 4.1. Since we are dealing with real-life problems, we need to have interpretable algorithms. In this way, domain experts can analyze in detail predictions made and understand real background reasons.

### 4.3. Evaluation and Interpretation of Results

As described in Section 2, the automotive company aims to reduce the number of vehicles called back, because stopping a vehicle on the road represents a delay in the drivers mission and economic damage. In the analytic perspective, the false positive identified by TETRAPAC must be reduced at the minimum while the true positive should reach the highest percentages. We can evaluate model performance through quantitative metrics, fully explained in [12]. The accuracy metric, widely used in the classification task on balanced datasets, is inappropriate here since we deal with unbalanced classes; thus, we focus on metrics that can be evaluated separately for each class, i.e., *precision*, *recall*, and their harmonic mean, i.e., the $F_1$-*macro* score, to have an overall performance indicator.

To better understand metrics details, we set as $P$ the class label corresponding to a truck with failure detection and as $N$ the label assigned to a vehicle without error. *Precision*, denoted as $p$, is the number of trucks correctly assigned to $P$ over the total number of

trucks predicted of class $P$ ($p = \frac{TP}{TP+FP}$). *Recall*, denoted as $r$, is the number of trucks correctly labeled as $P$ over the total number of trucks actually belonging to class $P$ ($r = \frac{TP}{TP+FN}$). $F_1$-*macro* is the harmonic mean of precision and recall:

$$F_1 - score = \frac{2 * p * r}{p + r} = \frac{2TP}{2TP + FP + FN}, \tag{1}$$

The main objective is to maximize the latter metric to reach 1. If precision and recall are pretty equal to 1, it means that the model predicts few false positives or false negatives. We also compute $F_1$-*macro* that aims to average overall performance overall classes. In practice, it is a simple mean of $F_1$-*score* for each category. It provides the same importance to all types even if they are unbalanced to understand overall model performance.

For regression tasks, as in case (2), the main objective is to maximize the quality of the predictive model by minimizing the model's error. Three metrics were selected: *mean squared error*, *coefficient of determination*, and *adjusted R-squared*.

*Mean squared error* (*MSE*) is the average of squared errors over all performed predictions, and it is defined as:

$$MSE = \frac{1}{n} \sum_i (y_i - \hat{y}_i)^2 \tag{2}$$

*MSE* tends to penalize the errors close to 0 less. It is always positive, and the smaller it is, the better the model.

*Coefficient of determination* ($R^2$), also known as R-squared, is defined as: $R^2 = 1 - \frac{MSE}{\sigma^2}$ and it represents the proportion of variance of $y$ explained by variation in $x$. R-squared assumes values in the range between 0 and 1. In particular, if $R^2 = 1$, the model perfectly explains data, while, if $R^2 = 0$, the model used does not explain data. It is important to observe that the increase in predictors corresponds to an increase in $R$-squared. However, it is not always true that as the complexity of the model increases, there is a better accuracy. For this reason, $R^2$ can be a misleading metric. *Adjusted R-squared* overcomes the previous limitation by incorporating the model's degree of freedom.

$$\overline{R}^2 = 1 - (1 - R^2)\frac{n - 1}{n - p - 1}; \tag{3}$$

where $p$ is the number of explanatory variables and $n$ is the number of samples. $\overline{R}^2$ also belongs to the range $(0, 1)$. In this way, we obtain a metric whose value will increase only if the accuracy of the model is improved.

The next relevant step consists of visualizing and interpreting the obtained results that are important to achieve the business aim. We analyze the trend of top predictors in three different ways. Through a scatter plot, we can visualize behavior in time of the selected feature. In this way, we can detect anomalies in values that can be the cause of failure. For classification tasks, we plot values assumed by top predictors divided in labels through an histogram plot to analyze differences between classes. For the regression problem, we first split data in ranges based on the battery state of health values and then visualize overall distribution through a box plot.

## 5. Experimental Results

Here we discuss a large number of experiments performed on actual telematics data with two objectives: (1) predicting a failure detection, described in Section 6, and (2) forecasting the battery state of health, introduced in Section 7.

In the considered use cases, telematics data and information from other data sources were obtained by the Azur DataLake (https://azure.microsoft.com/it-it/solutions/data-lake/ (accessed on 1 September 2021)). TETRAPAC has been developed in Azure DataBricks (https://docs.microsoft.com/it-it/azure/databricks/ (accessed on 1 September 2021)) using PySpark, Python [14], and SQL. In particular, the most exploited libraries for our work are: `Numpy`, `Matplotlib`, `Pandas`, `Scikit-learn`, and `Keras`.

During the development of the analysis pipeline, one crucial aspect to take into account is the tuning of hyperparameters. The resulting optimal values are the default ones validated with a `grid search` procedure.

Finally, we applied TETRAPAC to two different use cases. In the first one, shown in Section 6, the goal was to manage a predictive maintenance issue inherent to diagnostic trouble codes (DTC); while in the second one, described in Section 7, the purpose was to predict in advance the battery health status. The preprocessing phase (described in Sections 6.1 and 7.1) includes the same steps for both use cases to define specific KPIs depending on the analyzed scenario. Based on this, the best predictive model was built as illustrated in Sections 6.2 and 7.2. Further, as a validation of the proposed model, Sections 6.3 and 7.3 also discuss a state-of-the-art analytics pipeline based on deep learning algorithms.

## 6. Use Case 1: DTC

Here we discuss results inherent to DTC prediction. A DTC signal occurs when something goes wrong during vehicle travel. For example, trouble may occur at the engine level, with either a technical failure of sensors, or even extreme and dangerous health conditions of components. DTC is reported to the control room after a specific extended ignition time, and then the truck is called back to the workshop to solve the problem.

The goal is to define a mathematical model that can detect particular vehicle conditions before error appearance by analyzing vehicle health status and the adopted driving style. Then it will predict the occurrence of DTC in advance to optimize time and costs.

### 6.1. Data Preprocessing

Our dataset structure is not so clear and intuitive at first sight. For each mission achieved by the trucks, the telematics system records many features describing the driving style and the engine status. The key is finding an easy way to read all the variables using the collected values, for example pivoting available data. We remove outliers and noise during this phase, and we pay attention to information related to time, date, and kilometers to obtain coherent data in the final output.

To obtain the best results from the predictive model, we have to structure our data precisely. We exploit cases and controls setting to distinguish vehicles with DTC from vehicles without failures. In particular, only a few more than one hundred trucks show the error under analysis that monitors general battery status. We match controls to cases selecting the same telematics parameters in the same input window to obtain comparable data, as shown in Figure 4, respecting a ratio pair to 1:10. We define the *target variable* of the predictive model: cases will have a label equal to 1; controls will have one similar to 0.

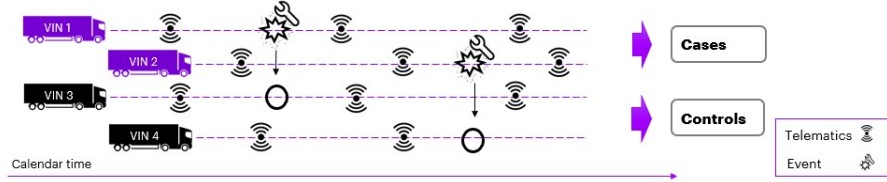

**Figure 4.** Case-control setting.

Then, we perform the feature creation step. Some instants are relevant for vehicle life until DTC ignition because they divide it into two different periods. Then, two temporal features (measured in days) are defined as shown in Figure 5: *time in stock*, i.e., the time passed in a factory until the vehicle is sold; *life before failure*, i.e., days of life of the vehicle under the responsibility of the customer before first failure detection. The latter is helpful to understand what happened to vehicles in the time window preceding the failure occurrence.

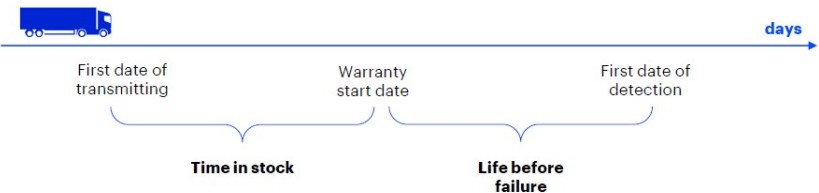

**Figure 5.** Relevant periods of truck life.

Then we select data related to defined days before the failure. We fix the input window equal to 30 days, considering a blind window of 2 days. These values have been chosen to evaluate different aspects, considering business needs and data availability. Then, we aggregate data over the input window, computing statistics and metrics to characterize time-series trends. In the end, we obtain 200 features starting from above 25 variables monitored by telematics. In particular, we show a discrete time-series example of a variable monitored by telematics in Figure 6.

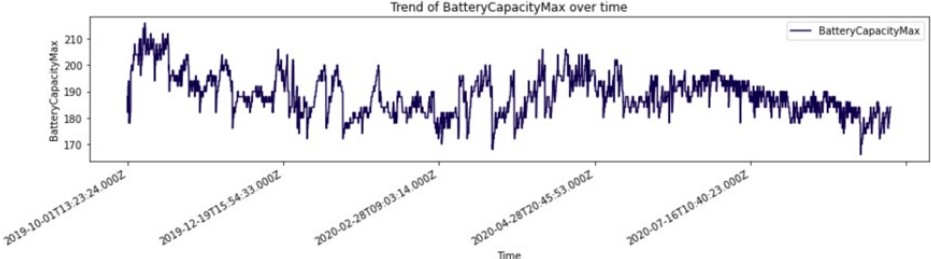

**Figure 6.** Time series plot.

We are dealing with a considerable number of features, but it is important to keep only relevant ones. We analyze the correlation matrix, shown in Figure 7, to remove highly related variables among them and ones poorly related to the target. We obtain only 54% of original features, above two hundred. Moreover, aggregation over time reduces dataset granularity: from more than ten million rows related to 7000 trucks sending data every mission (8 per day on average), we obtain one row per vehicle.

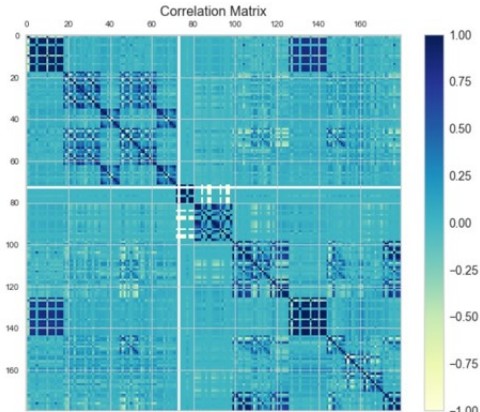

**Figure 7.** Correlation matrix.

Moreover, to understand which features influence the most DTC ignition, we exploit the Recursive Feature Elimination technique that provides in the output the optimal number of features to be selected with the relative optimal score for the model used. In Figure 8, we show its performance: it shows that only three variables are sufficient to obtain an $F_1$-*macro* score equal to 0.77.

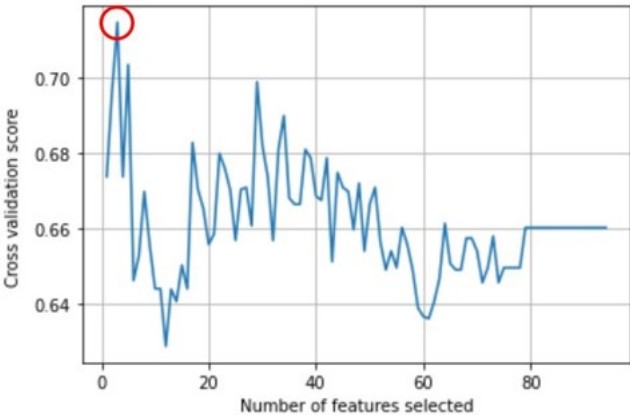

**Figure 8.** Recursive feature elimination performance with XGBoost.

### 6.2. Predictive Analytics and Interpretation of Results

The optimal model selected in the experimental phase is XGBoost Classifier, able to detect in advance vehicles with failure labeling them with a predicted target equal to 1, with an $F_1$-*macro* score equal to 0.77. We show in Figure 9 the best model performance.

| | XGBoost | | | Random Forest | | |
|---|---|---|---|---|---|---|
| | Precision | Recall | F1-score | Precision | Recall | F1-score |
| Class 1 | 0.77 | 0.46 | 0.77 | 0.86 | 0.40 | 0.68 |
| Class 0 | 0.94 | 0.98 | | 0.94 | 0.99 | |

**Figure 9.** Comparison of result through different metrics.

We computed the precision and recall of both classes, and then we evaluated the overall performance using $F_1$-*macro*. We note that since the dataset is highly unbalanced, metrics of the majority class (label equal to 0) are better than ones relative to the minority class (label equal to 1). Moreover, there are issues regarding data quantity and quality that significantly affect model performance. To improve precision and recall of the minority class, the one we are interested in predicting, we would need more DTC occurrences to have a more balanced dataset. On the other hand, in the future, the telematics system will be improved by adding more sensors to obtain even more precise modeling and then enhance the data-monitoring phase.

To understand the leading causes of error ignition, we visualize the behavior of model top predictors. In Figure 10, we plot behavior in the time of a variable monitoring battery charge for cases, namely vehicles with DTC. We note that there are values out of the standard range, so maybe this could cause the occurrence. To validate our hypothesis, we compare behavior between cases and controls. We see that these last ones mainly present values in the correct range (0–100), while there are a consistent number of cases (blue bars in Figure 11) having abnormal battery charge value around 160.

Despite numerous problems of data quantity and quality, model performance can be considered quite good. As an example for the monitoring battery charge level, top predictor behavior confirms and explains, in an exhaustive way, the model predictions. Finally, this aspect makes our model highly interpretable, and this is one of the fundamental points in business application.

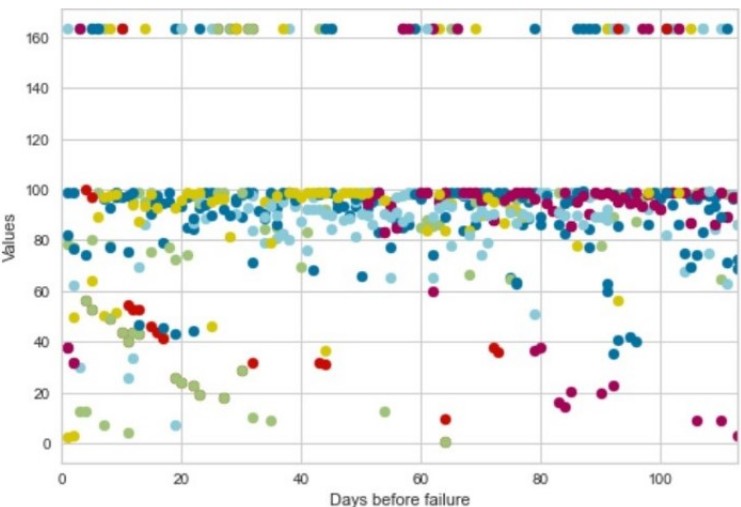

**Figure 10.** Behavior in time for cases.

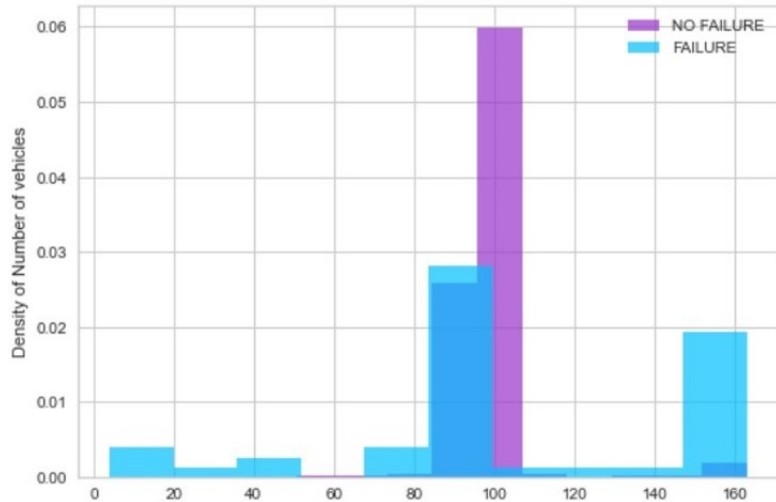

**Figure 11.** Battery charge histogram.

### 6.3. Comparison with Neural Networks

We now compare the proposed methodology with a *deep learning* algorithm. We selected the *recurrent neural networks* [15], tailored to time-series analysis. They allow processing sequential data to continue evolving throughout the duration. Among all available types of recurrent neural networks, *long short-term memory* (LSTMs) networks [16] are especially appealing to the predictive maintenance domain since their architecture is customized to learn from sequences. Even if feedback loops in the recurrent layer allow RNNs to keep the information in "memory" over time, it can be difficult to train standard RNNs to solve problems that require learning long-term temporal dependencies. This is because the gradient of the loss function decays exponentially with time, known as the vanishing gradient problem. Since LSTM units include a "memory cell" able to maintain information in memory for long periods, the vanishing gradient problem is solved; therefore, to the best of our knowledge, LSTMs are the best state-of-the-art neural networks to address a time-dependent predictive maintenance problem such as the one described in this paper.

We compared the two different approaches. On the one hand, for TETRAPAC , we consider datasets handled and manipulated by ourselves, as described above. On the other hand, using neural networks, we need to consider the initial granularity of the original dataset without any feature engineering and aggregation steps. In this way, we test if algorithms chosen in the data-driven methodology are optimal and if our data preparation

is adequate; however, in both approaches, to legitimize the comparison, we impose that the prediction needs to be in the future after the blind window. Ultimately, to obtain the best configuration of the network, we perform the tuning of hyperparameters.

So, to test the TETRAPAC methodology applied to this use case, we set case-controls with a ratio equal to 1:3 due to the higher imbalance between the two classes of vehicles. Moreover, we note that we must select the same variables monitored by telematics with the same values for input and blind windows to maintain this approach compared with the first one.

After the network configuration, we analyze model performance, as shown in Figure 12. To select the best model, we consider the $F_1$-*macro* as a more reliable metric. Based on the quantitative metrics, the NN-based approach gets the worst performance because of a low amount of data and a high imbalance between the two classes.

| | XGBoost | | | Random Forest | | | Neural Network | | |
|---|---|---|---|---|---|---|---|---|---|
| | Precision | Recall | F1-score | Precision | Recall | F1-score | Precision | Recall | F1-score |
| Class 1 | 0.77 | 0.46 | 0.77 | 0.86 | 0.40 | 0.68 | 1.00 | 0.14 | 0.58 |
| Class 0 | 0.94 | 0.98 | | 0.94 | 0.99 | | 0.86 | 1.00 | |

**Figure 12.** Comparison of NN results through different metrics.

In the end, we conclude that the XGBoost classifier leads better performance than LSTM, comparing the above results with the ones in Figure 9, confirming that TETRAPAC is more suitable for predictive maintenance issues.

## 7. Use Case 2: Battery

Here we discuss results relative to battery state of health (SOH). In fact, this element plays an extremely important role in non-electric vehicles. Many sensors such as headlights, infotainment systems, fridges, air-conditioning, and much more operate only thanks to energy produced by the battery. As a result, traveling with a degraded battery brings many complications until vehicle breakdown, this leads to a huge economic loss because users are forced to interrupt their work.

The goal was to create a model of earlier prediction to prevent battery breakage. The forecast model considered was based on temporal aggregation of input data, and battery health prediction was relative to the value after a predefined time horizon. In this way, during the blind window, the user will be able to take action to avoid a sudden battery breakdown.

### 7.1. Data Preprocessing

As performed for the first use case, in Section 6, we start with data manipulations steps. In particular, it is essential to structure the dataset to suit the forecasting model we want to build. In fact, as shown in Figure 13, it will be necessary to relativize time in order to be able to compare trucks at the same period of life.

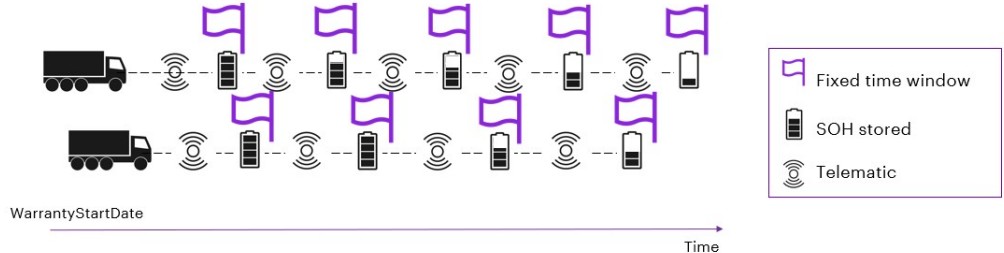

**Figure 13.** Time window settings.

In this context, features created concern both general health conditions and battery state of charge (SOC). In fact, on the one hand, we monitor the temperature range to which

the vehicle is subjected, the number of days it has been in stock and battery health status at the time truck was sold. Specifically, Figure 14 shows the behavior of this last KPI: trucks starting to circulate with a sub-optimal battery will have a predicted a health status in the low range. On the other hand, we have created KPIs related to battery SOC, such as the number of charges and discharges.

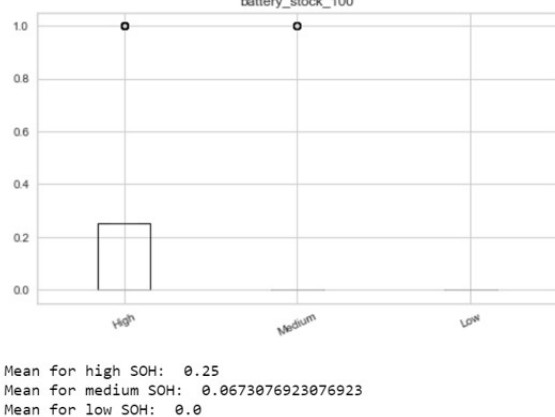

Mean for high SOH:  0.25
Mean for medium SOH:  0.0673076923076923
Mean for low SOH:  0.0

**Figure 14.** KPI behavior for trucks classified according to their battery health status.

Data are ready to be aggregated over time through statistics and metrics of interest. Starting with a set of 200 variables from telematics, we obtain a final set consisting of above 1500 features.

The choice of optimal input and blind window size is the result of several data-driven tests. The best scenario is to aggregate data over 1 month and make the prediction after 15 days.

After the creation of new parameters and time aggregation, it is important to be able to select those that affect battery SOH. This can be achieved by studying the correlation between KPIs. Obviously, we observe that parameters that measure specific aspects of the battery status are not deleted, such as voltage, temperature, charge, and capacity.

After this manipulation, we obtain above 150 features. As in the previous use case, this allows us to reduce data granularity: we deal with one row per vehicle.

Then we proceed by applying the recursive feature elimination, RFE, so we have 16 top predictors selected from the variables filtered by correlation analysis.

### 7.2. Predictive Analytics and Interpretation of Results

All algorithms integrated in TETRAPAC were tested with the 16 top predictors extracted. The best performing was the random forest regressor, as shown in Figure 15.

| XGBoost | | Random Forest | |
|---|---|---|---|
| R2_adj | MSE | R2_adj | MSE |
| 0.87 | 5.11 | 0.90 | 3.86 |

**Figure 15.** Comparison of results through different metrics.

This is a very good result and it makes real-world model applicability really promising. This last aspect is not only influenced by high values in metrics but is also based on easy model interpretability. This, as in the first use case, can be achieved analyzing the top predictors' behavior extracted by the best model. In particular, in Figure 16 we show time trend (a) and distribution (b) of battery capacity for vehicles split according to predicted battery health status: a degraded battery will have low capacity values.

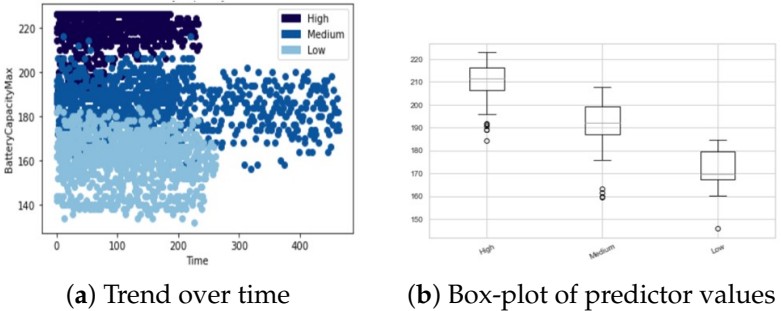

(**a**) Trend over time        (**b**) Box-plot of predictor values

**Figure 16.** Top predictor: battery capacity for different SOH ranges.

In addition, in Figure 17, we show the behavior of another top predictor in order to clarify model functioning. We note that to obtain a non-degraded battery result, we have high values of these predictors, proved by trend over time in (a), concentrated within a limited range, as shown by standard deviation box-plot in (b).

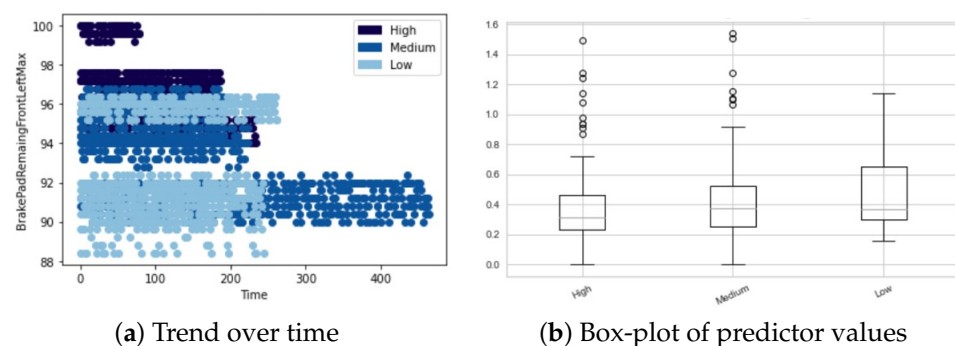

(**a**) Trend over time        (**b**) Box-plot of predictor values

**Figure 17.** Top predictor: brake pad remaining for different SOH ranges.

By analyzing all 16 top predictors, we note that battery capacity seems to fully explain the target variable. To better understand the underlying causes of this scenario, as shown in Figure 18, we train the model using different combinations of top predictors.

We can conclude that our model reaches the best performance with the entire set of top predictors. In fact, battery capacity only explains a partial behavior of the target variable.

In conclusion, following the TETRAPAC methodology, we offer an exhaustive predictive model that performs well and is quite interpretable.

| Model | MSE | R2 |
|---|---|---|
| Random Forest -16 top predictors- | 3.86 | 0.90 |
| Random Forest -only Battery Capacity- | 12.76 | 0.68 |
| Random Forest -15 top predictors without Battery Capacity- | 4.52 | 0.89 |

**Figure 18.** Model training with different combination of top predictors.

### 7.3. Comparison with Neural Networks

As described in detail in Section 6.3, we now validate TETRAPAC in this use case.

After hyperparameters were set in the optimal configuration and standardizing the dataset, we applied the LSTM neural network to one truck at a time and monitored the trend of the resulting MSEs. To test the performance, we considered two different contexts: in version 1, the test set was composed of the last records of the selected vehicle, while in version 2, it was composed of a completely new truck.

Finally, as shown in Figure 19, we compared the performance obtained through these two approaches. We observed that the lowest MSE corresponds to the random forest regressor. This indicates how manipulation of data, specifically feature modeling and temporal aggregation, contributed to the model performance. For further proof of good results of TETRAPAC , it would be appropriate to retest models in the future when more historical data are available. Moreover, it cannot be ruled out that in the future, with greater availability of historical data, the performance of a neural network-based model will be much better than the results found now. Perhaps they could be better than the results obtained with TETRAPAC as well. It is difficult to make a prediction of how the performance of these models will vary in the future. The quality and quantity of the data greatly influence a predictive model. So, it is possible that by testing TETRAPAC in other scenarios, its performance will be worse than that of a deep learning approach.

| XGBoost | | Random Forest | | Neural Network | |
|---|---|---|---|---|---|
| R2_adj | MSE | R2_adj | MSE | R2_adj | MSE |
| 0.87 | 5.11 | 0.90 | 3.86 | 0.66 | 13.46 |

**Figure 19.** Comparison of results between different approaches.

## 8. Discussion

Through TETRAPAC methodology, based on a solid theoretical background and motivated by concrete business needs, we succeeded in bringing tangible benefits to the customer and the company. The fundamental block of every analysis and future development is the data collection enabled by telematics. In particular, applying TETRAPAC to DTC detection (see Section 6) and to the battery SOH (see Section 7), we can perform with success the entire process of predictive maintenance. Among various benefits, the more relevant are increased customer satisfaction and optimized fleet management for the company and final customer.

### 8.1. Economic Benefits

Prediction of failures and status of vehicle components allows to anticipate services to the vehicle and to optimize the maintenance. Two possible scenarios for tangible economic benefits can be drawn.

**Scenario A: a failure is properly anticipated.**

We confirm the bold, to more effectively identify the two different scenarios A and B
In case of failure properly anticipated leveraging TETRAPAC methodology, the overall economic saving for each vehicle is shown by the following formula:

$$economic\ savings = cost\ to\ recover\ truck\ on\ road\ +\ cost\ to\ replace\ component\ +\ cost\ to$$
$$repairother\ damaged\ components\ -\ (customer\ call\ to\ go\ back\ to\ workshop\ cost + cost\ to\ repair\ component),$$

where we suppose that

$$cost\ to\ repair\ component < cost\ to\ replace\ component.$$

This scenario could also lead to the introduction of flexible M&R contracts; for example, if the driver does not consider the alert of going to a workshop to check the status of a component and then the component breaks with consequent damage of other components, the company can refuse to pay their repair, even though they are under warranty service.

**Scenario B: replacement of component is avoided when it is not necessary.**

On the other hand, the predictive model could also be used to avoid situations where the component does not need to be replaced; so, the total saving will be equal to the undue cost to replace the component.

*8.2. TETRAPAC Benefits and Weakness*

Here we summarize the main positive features of TETRAPAC and its main limitations.

TETRAPAC *benefits.* This research mainly works with telematics data and is characterized by complementary aspects: it highlights the general-purpose structure of analysis and other customized pipeline steps. On one side, we can leverage TETRAPAC to address different use cases optimally. For example, to predict failure or state of health of components, some of the data preprocessing steps of our methodology, namely data cleaning and feature selection, can be precisely followed as they were designed. On the other hand, there are specific parts of TETRAPAC , such as feature modeling, time-window selection, predictive analytics, and final model evaluation, that should be defined and adapted to new needs. Indeed they must be performed according to business requirements and in line with the best performance achieved. So, whatever business needs for what concerns this kind of failure occurrences and their prediction in advance, the presented approach can offer a reliable result. TETRAPAC is a predictive maintenance framework that could be easily adapted to any use cases dealing with telematics data.

TETRAPAC is entirely integrated into the Azure DataBricks service. In this way, the analytics process is end-to-end monitored and updated. This enables a future real-time validation to be implemented and integrated into Azure DataFactory. The purpose will be to compare predictive results with actual values based on our data available for real-time assessment of about 5000 vehicles per day.

TETRAPAC weaknesses

Although we said that TETRAPAC is a general-purpose methodology, the feature modeling step can not be fully automated. This phase must be modeled according to the final goals, with specific and constant support from domain experts.

Deepening the weakness analysis of each single-use case, we highlight the presence of high unbalanced classes for DTC detection and rare low values for SOH prediction. For the first problem, the model will tend to have a high percentage of false negatives. A solution may be the application of majority class undersampling or minority class oversampling, or a combination of these techniques. For the second problem, since vehicles under analysis have short life history, their battery is not degraded: consequently, the model will tend to predict higher SOH values than real ones. This can be fixed with regularization techniques to assign greater weight to vehicles with degraded batteries.

*8.3. Open Issues and Future Research Directions*

We will continue to increase telematics data monetization to address innovative and challenging real-life problems through data-driven methodologies. Future directions of this research work include: (i) the integration into TETRAPAC  of anomaly-detection techniques exploiting scalable one-class classifiers; (ii) the extension of TETRAPAC  with concept-drift detection techniques to automatically detect when the model is no longer performed correctly since data distribution of the new unseen data differs significantly from training data; (iii) the exploitation of TETRAPAC in similar application contexts. For example, TETRAPAC usage in the battery area of application, based on thermal vehicles, can be spread to the new electric context. In particular, among concerning points, there is electric vehicles' autonomy, charging time and cost, and battery maintenance cost. Some basic steps could be to develop a dynamic configuration, thanks to which the user can be constantly informed about battery SOH, charge optimization for health status management, and, as regards battery maintenance costs, it will be possible to increase battery life and to improve fleet management.

**Author Contributions:** Data curation, P.B., S.C. (Sara Cavaglion), S.C. (Sofia Cricelli), and E.L.; Methodology, P.B., S.C. (Sara Cavaglion), S.C. (Sofia Cricelli), E.L., E.M., and T.C.; Software, P.B., S.C. (Sara Cavaglion), S.C. (Sofia Cricelli), and E.L.; Validation, P.B., S.C. (Sara Cavaglion), S.C. (Sofia Cricelli), E.L., S.G., and T.C.; Writing—original draft, P.B., S.C. (Sara Cavaglion), S.C. (Sofia Cricelli), and E.L.; Writing—review and editing E.M., S.S., A.R., F.C., S.G. and T.C. All authors have read and agreed to the published version of the manuscript.

**Funding:** This work has been partially funded by the EU under the H2020 EnABLES project, Grant Agreement n. 730957.

**Institutional Review Board Statement:** Not applicable.

**Informed Consent Statement:** Not applicable.

**Data Availability Statement:** 3rd Party Data. Restrictions apply to the availability of the data used in this paper.

**Acknowledgments:** Thanks to Valentina Diaferio (Accenture S.p.A.) for taking care of the graphic part of Figure 1.

**Conflicts of Interest:** The authors declare no conflict of interest.

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
