# Peer review of "Empowering Commercial Vehicles through Data-Driven Methodologies"

_electronics, doi:10.3390/electronics10192381_

Round 1
Reviewer 1 Report
My main comment to the author is regarding the amount of data used to train the neural network (NN). The performance of neural network is very dependent on
- The network design
- the amount of data used to train the network.
Since the authors showed that there results are better than neural network, my question is also twofold
- Can a better neural network be designed for the application(s) in question: DTC or battery life prediction ( a different network for each case)?
- If we have more data say 1000x more data, will the NN outperform the TETRAPAC?
These are difficult questions. So a concrete answers may not be possible. However, a discussion on these topics will make the paper more complete.
Author Response
We thank the reviewer for the interesting observation, which gives us the chance to better explain and clarify these important aspects.
Regarding the first question, we choose Recurrent Neural Networks because they are the most suitable for temporal data to the best of our knowledge. Indeed, DTC and battery life prediction are both temporal data. Specifically, we choose LSTMs because are the most powerful RNNs and are able to learn long time relationship. In the revised paper, we better clarify this choice by adding a discussion in Section 6.3.
Instead, regarding the second question, we agree that the NN could outperform TETRAPAC with more data. However, it is difficult to estimate the order of magnitude of data needed to make this happen. Thus, we contextualize our results with the available amount of data. In the revised paper, we added a discussion on this second aspect in Section 7.3.
Reviewer 2 Report
Please, more attention to English language.
Author Response
We thank the reviewer for the comment, which gives us the chance to make the paper clearer. In the revised paper, we corrected some typos and English formulations.